# Vaccination Schedules Recommended by the Centers for Disease Control and Prevention: From Human-Readable to Machine-Processable

**DOI:** 10.3390/vaccines13050437

**Published:** 2025-04-22

**Authors:** Xia Jing, Hua Min, Yang Gong, Mytchell A. Ernst, Aneesa Weaver, Chloe Crozier, David Robinson, Dean F. Sittig, Paul G. Biondich, Samuil Orlioglu, Akash Shanmugan Boobalan, Kojo Abanyie, Richard D. Boyce, Adam Wright, Christian Nøhr, Timothy D. Law, Arild Faxvaag, Lior Rennert, Ronald W. Gimbel

**Affiliations:** 1Department of Public Health, College of Behavioral, Social and Health Sciences, Clemson University, Clemson, SC 29634, USAliorr@clemson.edu (L.R.); rgimbel@clemson.edu (R.W.G.); 2Department of Health Administration and Policy, College of Public Health, George Mason University, Fairfax, VA 22030, USA; hmin3@gmu.edu; 3Department of Clinical and Health Informatics, McWilliams School of Biomedical Informatics, The University of Texas Health Sciences Center at Houston, Houston, TX 77030, USA; yang.gong@uth.tmc.edu (Y.G.); dean.f.sittig@uth.tmc.edu (D.F.S.); 4School of Computing, College of Engineering, Computing and Applied Sciences, Clemson University, Clemson, SC 29634, USA; cscrozi@clemson.edu (C.C.);; 5Independent Consultant, Cumbria CA13 0RU, UK; 6Department of Pediatrics, School of Medicine, Indiana University, Indianapolis, IN 46202, USA; pbiondic@regenstrief.org; 7Global Health Informatics, Regenstrief Institute, Indianapolis, IN 46202, USA; 8Department of Biomedical Informatics, School of Medicine, University of Pittsburgh, Pittsburgh, PA 15206, USArdb20@pitt.edu (R.D.B.); 9Department of Medicine, School of Medicine, Vanderbilt University, Nashville, TN 37232, USA; adam.wright@vumc.org; 10Department of Sustainability and Planning, The Technical Faculty of IT and Design, Danish Center for Health Informatics, Aalborg University, 9000 Aalborg, Denmark; cgn@plan.aau.dk; 11Ohio Musculoskeletal and Neurologic Institute, Ohio University, Athens, OH 45701, USA; timothy.law@highmark.com; 12Department of Neuromedicine and Movement Science, School of Medicine, Norwegian University of Science and Technology, 7491 Trondheim, Norway; arild.faxvaag@ntnu.no

**Keywords:** vaccination schedules, Centers for Disease Control and Prevention (CDC), vaccination rules, clinical decision support systems (CDSS), machine-processable, CDSS rules, clinical quality language (CQL)

## Abstract

Background: Reusable, machine-processable clinical decision support system (CDSS) rules have not been widely achieved in the medical informatics field. This study introduces the process, results, challenges faced, and lessons learned while converting the United States of America Centers for Disease Control and Prevention (CDC)-recommended immunization schedules (2022) to machine-processable CDSS rules. Methods: We converted the vaccination schedules into tabular, charts, MS Excel, and clinical quality language (CQL) formats. The CQL format can be automatically converted to a machine-processable format using existing tools. Therefore, it was regarded as a machine-processable format. The results were reviewed, verified, and tested. Results: We have developed 465 rules for 19 vaccines in 13 categories, and we have shared the rules via GitHub to make them publicly available. We used cross-review and cross-checking to validate the CDSS rules in tabular and chart formats. The CQL files were tested for syntax and logic with hypothetical patient HL7 FHIR resources. Our rules can be reused and shared by the health IT industry, CDSS developers, medical informatics educators, or clinical care institutions. The unique contributions of our work are twofold: (1) we created ontology-based, machine-processable, and reusable immunization recommendation rules, and (2) we created and shared multiple formats of immunization recommendation rules publicly which can be a valuable resource for medical and medical informatics communities. Conclusions: These CDSS rules can be important contributions to informatics communities, reducing redundant efforts, which is particularly significant in resource-limited settings. Despite the maturity and concise presentation of the CDC recommendations, careful attention and multiple layers of verification and review are necessary to ensure accurate conversion. The publicly shared CDSS rules can also be used for health and biomedical informatics education and training purposes.

## 1. Introduction

Clinical decision support systems (CDSS), a key component of electronic health record (EHR) systems, have been adopted and used regularly in healthcare delivery in the United States of America and worldwide [1,2]. The effectiveness of CDSS has been demonstrated in multiple settings, particularly in enhancing preventive service orders and medication prescriptions [3,4]. An analysis of the Centers for Disease Control and Prevention (CDC)-collected, national 2015 data indicates that CDSS usage has reached 68–100% in primary care settings among practices with an EHR [5]. Most operational CDSS are rule-based systems; that is, the rules dictate the behaviors of CDSS [6,7,8,9]. Despite technological advances, the vision of sharable, reusable, and portable CDSS rules has not yet come to fruition. This reality presents at least twofold challenges: (1) CDSS rules have to be operationalized and maintained again and again in each organization, despite the fact that the sources of the rules may be the same. (2) This process of development and maintenance of CDSS rules can be a significant barrier for smaller practices, which may not have sufficient resources (e.g., professional IT personnel) as large medical centers. Lack of needed support subsequently may lead to the use of outdated rules in practice; eventually, patients would be negatively affected by the outdated rules. Therefore, keeping the CDSS rules updated and maintained in a timely manner is as critical as adopting the CDSS or adopting the clinical practice guidelines and professional recommendations. Keeping CDSS rules updated completes the rule life cycle. Updated and maintained CDSS rules also provide a foundation for clinicians to use the most updated rules in clinical practice to ultimately benefit patients. The development of reusable and sharable CDSS rules is a way to mitigate this challenge.

Figure 1 illustrates the conceptual use of a CDSS in healthcare delivery and the CDSS’s role in the cycle of the learning health system [10], which refers to an iterative process where data from clinical practice are continuously collected, analyzed, and applied to enhance clinical decision-making and overall system efficiency. Figure 2 presents a conceptual framework for a typical rule-based CDSS system, including typical inputs and outputs related to the CDSS. Here, we use CDSS rules to refer to the executive rules proposed by the AHRQ CDS Connect community [11].

Ontology [12,13] is an enabling technology to achieve the semantic web goals; that is, in the semantic web era, both data and semantics will be portable [14,15]. Tom Gruber defined ontology as: “An ontology is a specification of a conceptualization” [16]. A biomedical ontology is a computable representation of knowledge from one or more biomedical domains and typically includes entities and relationships [13]. Biomedical ontologies have a strong commitment to biomedical realism and logical grounding [17]. One of the main use cases for constructing ontologies has been to facilitate information retrieval and interoperability. Bioportal is a primary directory of ontologies in the biomedical field [18]. We constructed a CDSS ontology to facilitate the development and maintenance of CDSS rules [9,19,20]. Leveraging ontologies to represent machine-processable CDSS rules to realize sharable, reusable, and interoperable knowledge artifacts can be valuable resources for the health IT and medical informatics fields [19].

Standards have been widely used in healthcare delivery for various purposes, particularly in sharing patient information between organizations or healthcare systems. Fast Healthcare Interoperability Resources (FHIR) is a relatively new standards framework developed by HL7, an international standards development organization [21,22]. FHIR incorporated many years of standards development and implementation experience, leveraged advantages, and overcame disadvantages from many types of prior standards [22]. FHIR emphasizes implementation and interoperability and leverages the latest standards and available technologies. It has gained more and more attention and has been adopted broadly in the healthcare industry for various purposes [22].

Clinical quality language (CQL) is a language standard developed by HL7 that is intended to be human-readable [23,24]. It was developed for clinical quality measures and decision support artifacts, such as CDSS rules. CQL is used to represent and share clinical logic, and it can be converted to the Expression Logical Model (ELM) format, which is a machine-processable format [23,24]. CQL was developed initially to facilitate clinical care quality measures for Medicare and Medicaid programs. Later, it evolved into a medical logic language for CDSS artifact development.

Our group strives for reusable and sharable CDSS rules for primary care settings, particularly in resource-limited environments. We used CDC-recommended vaccination schedules [25] as the source of the CDSS rules. In this paper, we introduce the processes undertaken, the results achieved, the challenges encountered, and the lessons learned when converting the CDC-recommended vaccination schedules [25] from human-comprehensible guidelines into machine-processable CDSS rules. We hope our process can guide peers in preparing their CDSS rules and that the results (i.e., different formats of CDSS rules on vaccinations) can be used/reused by peers as HealthIT artifacts or educational materials. Preparing CDSS rules sets the stage for CDSS rule management and maintenance; both steps consist of the essential components for CDSS rules’ life cycle.

## 2. Methods

### 2.1. Rules Conversion Workflow

The development of the CDSS vaccination rule (≤18 years old) went through three stages—piloting, formal development, and conversion stages. During the piloting stage, we used measles, mumps, and rubella (MMR) and human papillomavirus (HPV) as two examples to explore the steps that should be followed and principles that should be adhered to during the development and review of all vaccination rules. We followed the same steps in each stage for each format, including analyzing, developing, reviewing, revising, and testing. At the time, we included four format types: tabular, charts (an example will be shown in Section 3), MS Excel (Microsoft 365 Apps for enterprise), and CQL 1.5.3. We started with the recommendations for those younger than 18 years. Figure 3 presents the overall workflow of the vaccination rules’ development process. One author (XJ) started the conversion from the recommendation to tabular format, and two authors (HM and YG) independently reviewed and verified all the rules. All three authors have formal medical education.

In particular, we used the CDC-recommended vaccination schedules (≤18 years, 2022 version) as the knowledge source for the CDSS rules [25]. We first analyzed a specific vaccine (e.g., MMR) and all related conditions to the vaccine in the recommendation, which includes five tables, one appendix, and one note. For some vaccines, we also included extended publications cited in the CDC vaccination schedules to provide more detailed information. We then deliberately and explicitly expanded all conditions related to that vaccine and manually converted them into more granular rules, one by one, in a tabular format. We organized each vaccine in a single table, and each row in the table corresponded to a CDSS rule. The original recommended schedules were presented in a concise presentation intended to be used by healthcare professionals. The tabular format added and expanded all implicit recommendations into explicit, granular, and singular rules, which can be demonstrated by the number of final rules. The tabular format is intended for programmers to use. The remaining vaccines were then converted, reviewed, and verified by following the same steps and principles formulated via preparing MMR and HPV. All inaccurate conversions or representations, missed representations, and disagreements were discussed among the three authors to reach a consensus. The tabular format was prepared and organized in MS Word files (Microsoft 365 Apps for enterprise).

### 2.2. Specifications Related to Each Format

The tabular format was then used to generate charts, which were created through a piloting stage and a formal creation stage. One research assistant used the tabular format to generate MMR and HPV charts based on iterative discussions, revisions, and review meetings. Two research assistants then followed similar steps using tables, original schedules, and charts of MMR and HPV to create the charts for the rest of the vaccines. Each chart underwent numerous discussions, revisions, and cross-checks to verify its presentation and interpretation without ambiguity. The charts were created using a free online tool, diagrams [26], and later were exported into PNG and JPEG formats. The charts provide a complementary and more intuitive representation of the detailed vaccination rules.

The MS Excel format was developed in a very similar way to the tabular format, but with more granular fields. The MS Excel format is intended for programmers to develop, identify, and map parameters required during implementation, management, and maintenance of CDSS rules for rule updating or modification purposes. We used tabular format and original vaccination schedules to prepare the MS Excel format. All the rules for the same vaccine were organized into a single MS Excel file. The MS Excel format was used as a structured CDSS rule format to facilitate CDSS rule management and maintenance later.

The CQL format was developed in a mix of two ways: (1) We used the Agency for Healthcare Research and Quality (AHRQ) CDS Connect CQL rule authoring tool [11], an online CQL authoring tool to start CQL file creation. (2) We then used a text editor (e.g., Notepad++, V8.7.5) to further edit the CQL files based on the tabular format and the original CDC recommendations. The CQL files were tested for syntax errors with the CQL ELM converter [27]. Additionally, the CQL files were tested using our own unit-test framework with hypothetical patient records in HL7 FHIR resources (version 4.0.1) [28]. This unit-test framework is derived from previous efforts by mCODE [29]. Figure 4 presents a more detailed flow of the conversion.

### 2.3. Quality Control Strategies

In addition to cross-review and cross-verification, which were implemented during the development and conversion of the rules, we implemented several other strategies to improve the consistency of the work and ensure the quality of the CDSS rules.

During cross-verification and cross-review, we used Excel files to track the problems, questions, and confusion each of us noticed. We then went through the list during discussion meetings to resolve them one by one. Furthermore, we have a to-do list for each vaccine to ensure that all errors have been rectified.

We established the same charting rules to follow while creating the charts for the CDSS rules. For example, we used the same color-coding convention to create charts for all vaccines. Blue was used to describe a to-do list; green was used to describe conditions; red was used to describe contraindications; orange was used to represent follow-up visits; and purple was used to represent different vaccines. We used machine codes for each color to provide a consistent outlook for the created charts. We aim to use consistent styles for different vaccines. Furthermore, for the chart format, we defined the usage of shapes, arrows, and lines to represent the same meaning between vaccines, which provided consistent meanings and interpretations for all vaccine charts to avoid confusion or ambiguity.

To test the CDSS rules, we have developed metrics based on each rule’s characteristics, patient data, vaccination conditions, and recommendations to prepare test cases, both positive (should trigger the CDSS rule) and negative (should not trigger the CDSS rule) cases. We then used different colors to distinguish the rules that had passed the test from those that were waiting to be tested, so as to better track the progress of the work.

We followed these steps during the vaccine rule analysis and rule development for each vaccine: Work on the normal schedule first. Examine the conditions before the vaccination. Examine the conditions after the vaccination. Examine the normal interval between dosages. Examine the abnormal interval. Examine the medical indications. Examine the special conditions and examine the contraindications.

## 3. Results

A total of 19 vaccines, organized into 13 categories, were converted, reviewed, and verified. The formats include tabular, chart, and MS Excel. The internal review and verification processes were completed for the tabular, charts, and MS Excel formats. The detailed rules for the distribution of the 13 categories are presented in Table 1. Currently, there are 465 rules for 19 vaccines. We successfully developed and tested 12 CQL rules/files, and the development of the remaining vaccine rules into CQL format is ongoing. Testing CQL file syntax was not a one-step success, and the process involved numerous failures, debugging, modifications, and retesting, all of which were iterated upon before all CQL files passed the syntax tests successfully. The patient FHIR resources that were used to test the CQL files were prepared based on the CDSS rules in Excel format. All CQL files were tested successfully with patient FHIRE resources, including positive and negative cases. The CDSS rules are shared through a GitHub repository [30] with the communities to encourage broad adoption, reusing, and sharing of the vaccination rules for clinical care, research, education, and training purposes. Table 2 presents the rules in various formats and the validation methods used for these rules.

Figure 5 illustrates a screenshot of the CDC-recommended vaccination schedule (2022 version’s Table 1 [25]), which is the primary information source we used to develop the CDSS rules. Figure 6, Figure 7, Figure 8 and Figure 9 show the CDSS rules for MMR in tabular (Figure 6), MS Excel (Figure 7), CQL (Figure 8), and chart (Figure 9) formats. Figure 10 illustrates the anticipated use of these CDSS rules, a central repository, and rule repository instances for each use case. We expect the central vaccination rules repository to be updated regularly, and each rule repository instance can leverage the updated CDSS rules from the central repository. Furthermore, each rule repository instance can be customized and localized based on its use cases as needed.

## 4. Discussion

Our work contributes uniquely to both medical and medical informatics fields by converting vaccination schedules into machine-processable formats. The immunization schedules were converted into 263 rules in 1997 [6]. Although the main principles of immunization are similar, more definite rules can now be formulated two decades later. The change in the number of rules also reflects the rapid and continued advancement of medicine.

### 4.1. Significance of the Work

We highlight the significance of our work, both the process and outcomes, from several aspects.

First, our work contributes to clinical care and preventive care. The CDC-recommended vaccination schedules were used as the information source to develop the CDSS rules. The CDC-recommended vaccination schedules are broadly adopted and followed in clinical practice. Therefore, there is a large user base for the sources of the CDSS rules, regardless of format. Our work converted the schedules from human-readable to machine-processable formats, which expanded the use settings of the schedules to include the health-IT-developing community and vendors. Our work further amplified the effects of the CDC recommendations via additional channels and was extended to additional potential users beyond healthcare professionals. This is of particular significance considering the emergence of misinformation and doubts about truth and science in childhood vaccinations. Our work will reach even more healthcare providers over time.

A second significance of this study is related to the publicly accessible and multiple formats of the CDSS rules. This contribution is to the medical informatics field. The multiple formats of the CDSS rules can be directly used by CDSS developers, health IT vendors, medical informatics educators, and trainees, either in their CDSS products or during education or training activities, to understand the process, compare the results, and provide alternatives. Web-accessible open resources can serve as a central hub for CDSS rules in both industry (for CDSS products and CDSS operation) and academia (for education materials and research projects), from conceptualization to development, validation, and finalizing CDSS rules in different formats. Multiple CDSS rule formats present complementary possibilities to implement CDC-recommended vaccination schedules more broadly and, in additional formats, significantly enhance the effects of the schedules. The effects of reusable rules are particularly significant in resource-limited settings. The patient population served in such settings will ultimately benefit from the work.

The last contribution of the study is the machine-processable format of CDSS rules, which can significantly reduce the number of duplicate efforts to develop such rules in virtually every organization. For these CDC-recommended vaccination schedules, the machine-processable CDSS rules can be reused and shared across institution borders, which is a significant improvement from the current practice, in which each institution prepares its own set of CDSS rules. We hope the CQL format of the CDC-recommended vaccination schedules will have a positive impact on the health IT industry, enabling the implementation of CDSS rules across institutions. Although the current rules apply only to CDC-recommended vaccination schedules, the principles and processes can be applied to other types of CDSS rules beyond vaccination schedules.

Although making CDSS rules portable is a breakthrough in the medical informatics field, we must acknowledge that this work is based on the continuous advancement made in healthcare data standards’ development and adoption. Without FHIR resources, existing patient data in the FHIR resources format [22], as well as CQL, none of our work would have been feasible. Such efforts would not have been successful a decade ago, before the widespread adoption of FHIR and the invention of CQL. We just stand on the shoulders of other giants in the medical informatics field. In this study, we bypassed the well-documented curly brace problems [31,32,33] by leveraging HL7 FHIR resources. The CQL files were tested independently of any particular EMR or EHR systems. We used patients’ HL7 FHIR resources, which serve as a virtual layer between the underlying EMR/EHR systems and our CDSS rules. Therefore, no universal format of EHR schema is required as long as the parent data can be transformed into the FHIR resource format. In CQL rules, we used the CDSS Ontology and the coded value sets; most of the coded value sets are from the National Library of Medicine Value Set Authority Center. These steps increased the standardization of the CQL rules and paved the way for reusability and shareability of the CQL rules. We would like to emphasize the innovation of our project, even though we leveraged FHIR and CQL significantly.

As we mentioned earlier, our group is not the first group to work on vaccination schedules. Miller et al. explored the field and published their results successfully decades ago [6,7,8,34,35,36,37,38,39,40]. However, in the past two decades, we have experienced significant hardware and software advancements and have witnessed progress and success in data standard development, adoption, and implementation. Therefore, although our work is similar to Miller’s project, it focuses more on the reusability and portability of the CDSS rules. Notably, we learned numerous key points from their work and publications, which significantly reduced our trial-and-error efforts. Another related project is Immunization Calculation Engine (ICE) [11], which is an AHRQ-funded project that can provide recommendations for routine immunizations of particular patients. The primary goal of our work is similar, that is, to provide accurate immunization recommendations for patients. However, we approached the problem from different angles: ICE leveraged vMR 1.0 schema, which is a standard patient record information model, whereas our work leveraged FHIR resources and CQL. We focused only on CDSS rules, which are the core of the ICE engine, but not the only component of the ICE project. In addition, our CDSS rules are shared separately, including semi-structured (tabular), structured (MS Excel), executable rules (CQL), and human-readable charts, which is a unique contribution of our project.

### 4.2. Challenges and Limitations of the Work

Currently, all conversion efforts from human-readable to machine-processable formats are manual efforts. It is not ideal, but a reality. The primary bottleneck in manual efforts is most evident in the continuous updating stages. The vaccination schedules, similar to other clinical practice guidelines, are living documents. The recommendations will be updated when we reveal new knowledge in this particular field. It would be more difficult to scale up or update the manual work. If the different formats of the rules can be adopted broadly, the effort would be indeed worthwhile. However, promoting such resources to target users is as challenging as converting per se.

A much larger challenge is to increase potential users’ awareness of the existing CDSS rule resources. Although publications and conference presentations are routinely used to promote such resources, the results can be limited. Promoting current resources and increasing users’ awareness should be a long-term and continuous effort. The value of this work could be maximized with a larger scale of adoption; however, with limited adoption, its effects could be very limited—that is, the effects are eventually determined largely almost by the number of adoptions.

Systematic testing is ongoing. Tests are necessary, but often not explicitly or carefully planned. We realize that systematic testing can be a large project by itself. We must prepare carefully, purposefully curated patient records for positive and negative cases to ensure all logic is tested thoroughly. A platform to conduct the testing easily and in scale and a way to track the testing results and resolutions to the issues is necessary. We have conducted unit testing for the CQL rules; however, larger-scale systematic testing is an ongoing effort.

Additional challenges we faced in the project included the following: (1) incomplete immunization history of individual patients (e.g., some cases are not recorded in the immunization history systematically; other cases such information is not accessible at the point of care when needed, as noted by Miller too [38]); (2) incomplete history of patient’s conditions, medical diagnoses, or anaphylaxis status (similar to immunization records, incomplete information can be caused by lack of availability or accessibility at the point of care [38]); (3) nonstandard ways to represent patients’ immunization records, medical history, or healthcare providers’ procedures; (4) whether the interval (between doses) calculation should be in a fixed format (table) or performed when the program is executed; and (5) how to translate “age-appropriate complete vaccination of OCV7” during conversion.

The practical implications of these challenges suggest that converting the CDSS rules into a machine-processable format, despite being a long-standing challenge in the field, is one step closer to these rules being broadly adopted and ultimately benefiting patient care. Other steps, such as promotion, adoption, and implementation, are at different dimensions and require different expertise and experience; however, these steps are as important as the conversion step in order to fulfill the potential of these rules. Meanwhile, we have to acknowledge that, even if all conversions, promotions, adoptions, and implementations are conducted successfully, seamlessly, and perfectly, this will not guarantee positive outcomes during operation if we consider the very likely possibility of unavailable or inaccessible patient data needed to execute these rules successfully. Therefore, we have to recognize the differences between an ideal lab setting and day-to-day operations to execute these rules. We must also put the expectations within the right context of daily operation, with some tolerance or understanding if there are any failures.

### 4.3. Lessons Learned Through the Process

During the converting process, we noticed the following points that can be easily missed but are critical in converting the rules faithfully: (1) The rules must be based not only on the current schedules and tables but also on associated external documents; therefore, do pay attention to all citations and external links when using the schedules. (2) The annotations (e.g., the valid and invalid doses) to the tables are critical in converting the rules accurately and comprehensively. (3) Determining age thresholds accurately is important (e.g., ≥ versus >). (4) An unknown status should be explicitly represented if it is stated in the schedules. (5) Attention should be paid to inexplicit information in the schedules; they should be made explicit during conversion. (6) Multiple layers of verifications and validations are necessary to ensure the quality and accuracy of the rules.

### 4.4. Path to Conversion

We want to emphasize that although this paper presented all the steps as a linear process, the reality was far from linear. We performed numerous exploration and testing efforts based on existing technologies, our expertise, and available tools. A full paper like this one can never fully capture or demonstrate all the efforts.

We selected the CDC-recommended immunization schedules to develop reusable and sharable CDSS rules primarily because of their maturity, straightforward nature, and concise presentations. However, complexity, extended documentation, and representation challenges emerged when manually converting the rules. The current schedules are intended for use by healthcare providers and are not the most practical guide for programmers who require explicit, detailed, accurate, and complete instructions. Although the interoperability challenges [41], curly brace issues [32,42], and clinical and temporal complexity [6] are well recognized and documented in the medical/health informatics community, technical challenges are not the only ones faced when developing machine-processable rules.

## 5. Future Work

We are currently working on the systematic testing of the CQL rules. We hope to share the approaches, tools, and codes when we accomplish them. Such testing tools can be a critical component of the life cycle of CDSS rules. The CDSS rule management and maintenance work will be published separately. Another future direction is the addition of more formats of the CDSS rules, such as XML and JSON formats. We hope the publicly shared artifacts will help reduce the number of redundant efforts across borders since these recommendations have been broadly adopted. Meanwhile, we will provide more detailed documentation, along with this manuscript, so that future efforts on converting vaccination rules can be significantly more structured or diminished, ideally.

## 6. Conclusions

We have presented the process, results, challenges, and lessons learned in converting CDC-recommended vaccination schedules from human-readable formats to tabular, charts, MS Excel, and CQL formats and have shared the resources publicly as open resources. CQL is a machine-processable format that can be reused and shared easily. The CDSS rule resource, which is the result of our effort, can be used by the health IT industry, clinical care institutions or practices, and medical informatics educators as operational CDSS rules or educational materials for teaching and training. We hope that the process and experience we shared will guide future efforts to convert other types of CDSS rules into machine-processable formats. Our effort fills a gap in developing and sharing reusable, machine-processable CDSS rules, which is a critical step to promoting the sharing and reuse of health IT artifacts to save resources and maximize the benefits.

## Figures and Tables

**Figure 1 vaccines-13-00437-f001:**
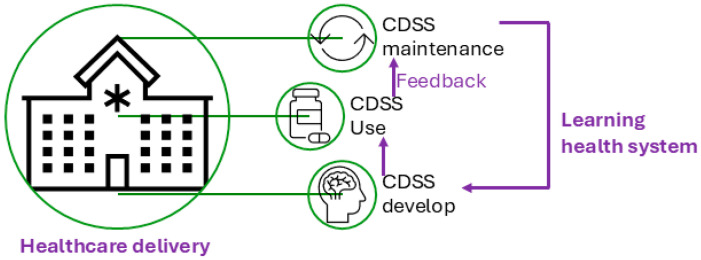
Clinical decision support system (CDSS) and its role in healthcare delivery.

**Figure 2 vaccines-13-00437-f002:**
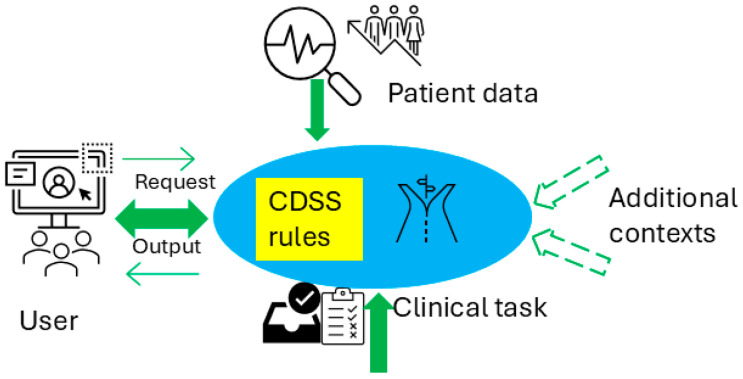
Conceptual architecture for a rule-based CDSS.

**Figure 3 vaccines-13-00437-f003:**
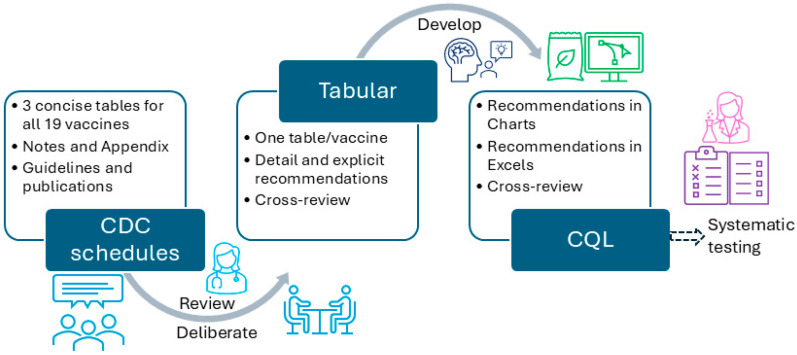
The overall workflow to convert CDC-recommended vaccination schedules to different formats.

**Figure 4 vaccines-13-00437-f004:**
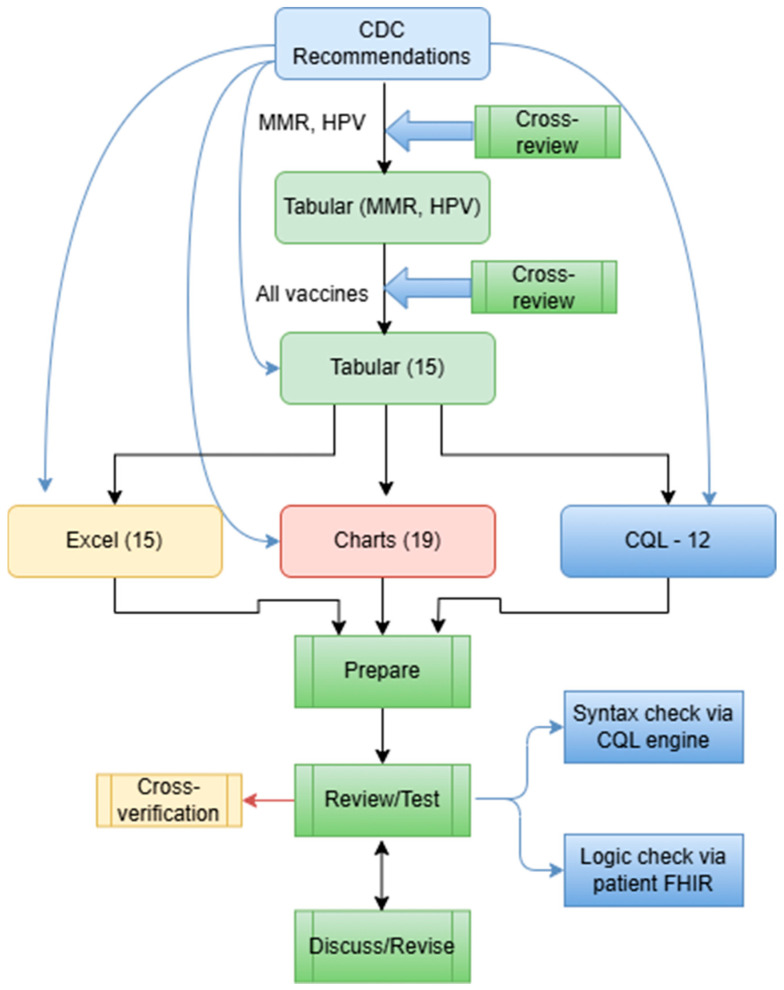
A more detailed workflow of rule conversion in various formats, including 15 Tabular files, 15 Excel files, 19 Charts, and 12 CQL files.

**Figure 5 vaccines-13-00437-f005:**
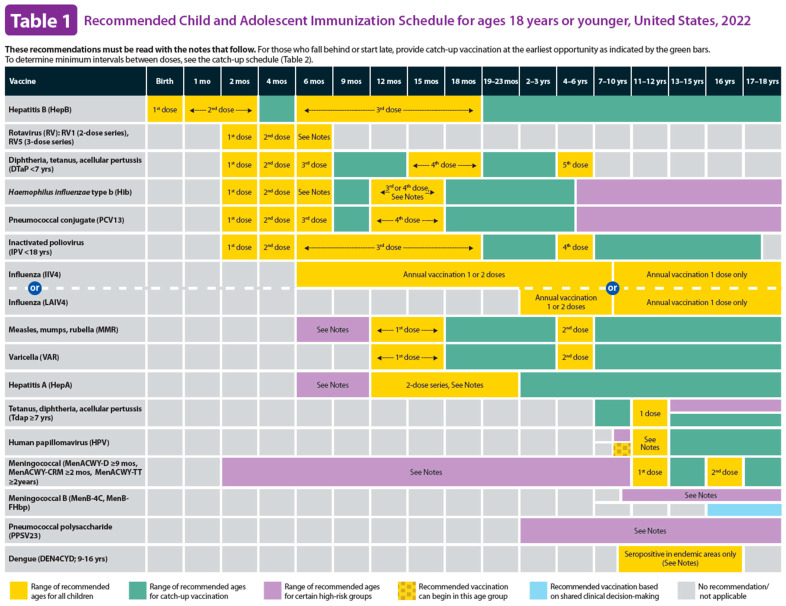
CDC-recommended vaccination schedules: Table 1 (2022 version)—the primary information source of CDSS rules [25].

**Figure 6 vaccines-13-00437-f006:**
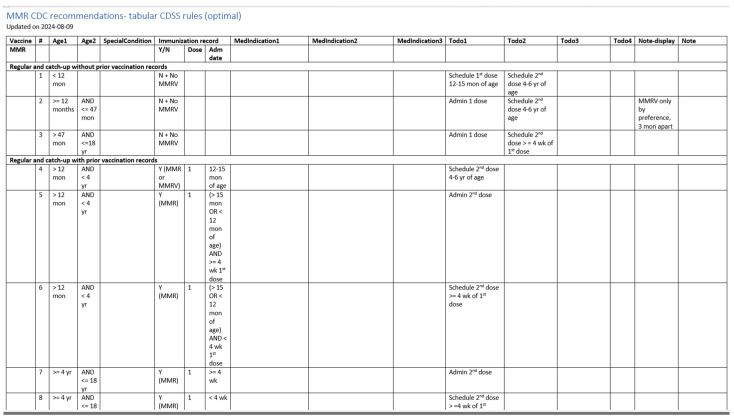
A screenshot of the CDSS rules in the tabular format for the MMR rules (partial).

**Figure 7 vaccines-13-00437-f007:**
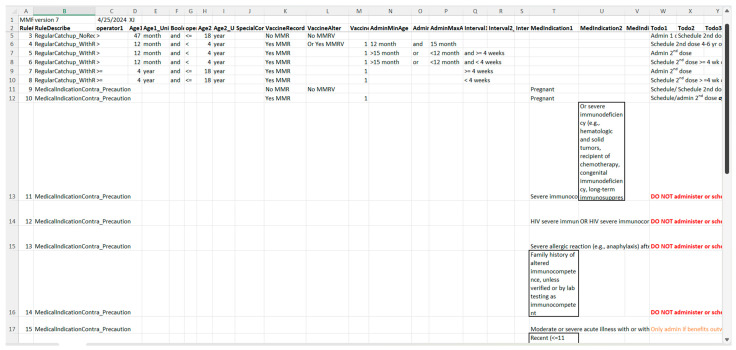
A screenshot of the CDSS rules in the MS Excel format for the MMR rules (partial).

**Figure 8 vaccines-13-00437-f008:**
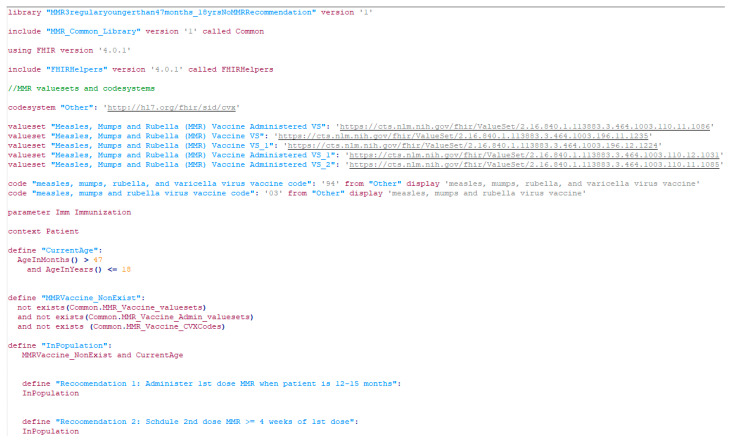
A screenshot of a CDSS rule in the CQL format for the MMR recommendation (partial).

**Figure 9 vaccines-13-00437-f009:**
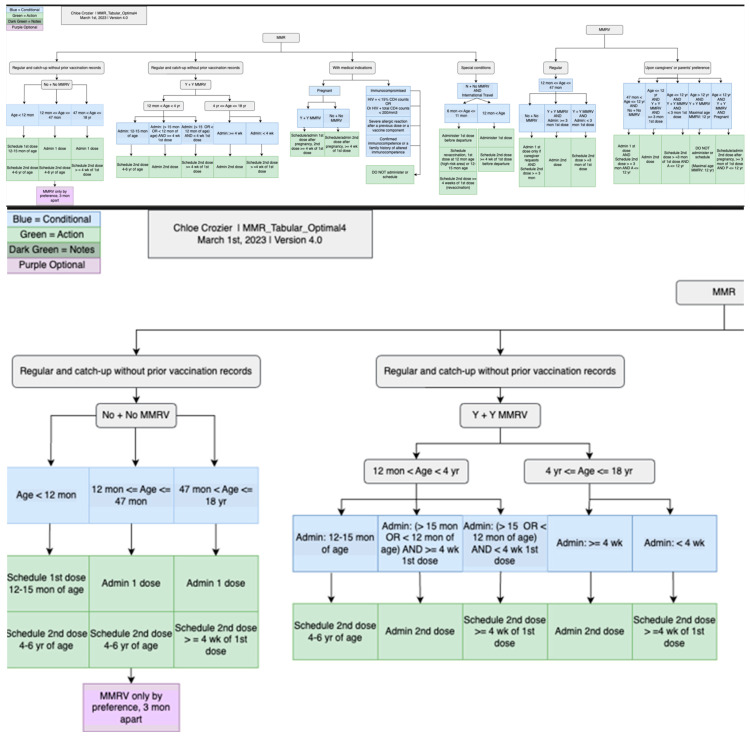
CDSS rules for MMR recommendations in chart format: an overview of the whole chart (**top**) and a partial chart to provide granular details (**lower**).

**Figure 10 vaccines-13-00437-f010:**
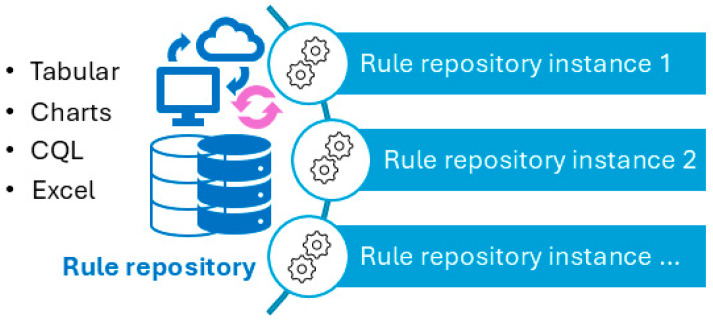
Usage of CDSS rules: central rule repository (centralized updating) and multiple instances (individual customization and localization can be made to meet individual use requirements).

**Table 1 vaccines-13-00437-t001:** Summary of the rules for the CDC-recommended immunization schedules (≤18 years, 2022).

Vaccine Category	Abbreviation	Regular or Catch-up	Medical Conditions /Special Situations	Contraindication /Precaution	Total
Dengue	DEN4CYD	2	-	7	9
Diphtheria, tetanus, and acellular pertussis	DTaP, DT	16	14	7	37
Haemophilus influenzae type b	Hib	16	20	4	40
Hepatitis A	HepA, Twinrix	6	2	4	12
Hepatitis B	HepB	8	14	3	25
Human papillomavirus	HPV	15	18	2	35
Influenza (inactivated and live and attenuated)	IIV4, LAIV4	13	-	33	46
Measles, mumps, and rubella	MMR, MMRV	16	2	10	28
Meningococcal serogroups A, C, W, and Y	MenACWY-D, MenACWY-CRM, MenACWY-TT	12	34	5	51
Meningococcal serogroup B	MenB-4C, MenB-FHbp	8	8	4	20
Pneumococcal 13-valent conjugate, 23-valent polysaccharide	PCV13, PPSV23	20	30	4	54
Poliovirus (inactivated)	IPV, tOPV	33	6	3	42
Rotavirus	RV1, RV5	11	-	8	19
Tetanus, diphtheria, and acellular pertussis	Tdap, Td	12	9	6	27
Varicella	VAR, MMRV	12	-	8	20

**Table 2 vaccines-13-00437-t002:** Summary results of rule formats and validation methods.

Rule Format	Number of Files	Number of Vaccines	Number of Rules	Validation Methods
Tabular	15	19	465	Cross-review and cross-verification
Excel	15	19	465	Cross-verification
Charts	19	19	465	Cross-verification
CQL	12	1	12	CQL ELM Converter (syntax) and CQL engine + patient FHIR resources (logic)

## Data Availability

The original contributions presented in this study are included in the article. Additional data are published in a GitHub repository [30]. Further inquiries can be directed to the corresponding author.

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
