# Peer review of "Vaccination Schedules Recommended by the Centers for Disease Control and Prevention: From Human-Readable to Machine-Processable"

_vaccines, 2025, doi:10.3390/vaccines13050437_

Round 1
Reviewer 1 Report
Comments and Suggestions for Authors
The article is devoted to an important and relevant topic related to the analysis and preparation of recommendations for vaccination schedules .
Authors must revise the article according to the following requirements.
- The authors should perform algorithmization for the proposed approach.
- Authors should pay attention to the issue of accounting for incomplete (censored) data and provide examples.
- In Table 1, the cell borders need to be adjusted so that all the text can be seen.
Comments on the Quality of English Language
The English could be improved to more clearly express the research.
Author Response
Reviewer 1
Comments and Suggestions for Authors
The article is devoted to an important and relevant topic related to the analysis and preparation of recommendations for vaccination schedules .
Authors must revise the article according to the following requirements.
- The authors should perform algorithmization for the proposed approach.
Response: Thank you for your suggestion. We have added one more figure, new figure 4, to explicitly lay out the steps in generating immunization recommendation rules.
- Authors should pay attention to the issue of accounting for incomplete (censored) data and provide examples.
Response: Thank you, we have added the following texts in the Discussion, section 4.2.
~~
Additional challenges we faced in the project: 1) incomplete immunization history of individual patients (e.g., some cases are not recorded the immunization history systematically, other cases such information is not accessible at the point of care when needed, noted by Miller too [38]); 2) incomplete history of patient’s conditions, medical diagnoses, or anaphylaxis status (similar to immunization records, incomplete information can be caused by lack of availability or accessibility at the point of care [38]);
~~
- In Table 1, the cell borders need to be adjusted so that all the text can be seen.
Response: Thank you, we have adjusted the table to make all content visible.
Reviewer 2 Report
Comments and Suggestions for Authors
1.The abstract provides a comprehensive overview of the research’s objectives, methods, and outcomes. However, it should compare itself with existing methods to highlight the advantages and improvements. (e.g., evaluation metrics: accuracy).
2.The innovation of this study should be emphasized more explicitly, especially how CQL and FHIR break limitations of traditional approaches in converting human-readable guidelines into machine-processable rules.
3.The title of Section 2.1 "Vaccination Schedule Rules Converting Workflow," could be more concise. You could change it to "Rule Conversion Workflow".
4.In the Data Presentation part, while Table 1 summarizes the distribution of rules, additional tables could enhance the description of data collection and preprocessing. For instance, a table outlining the preprocessing standards for different formats (tabular, CQL, Excel) and validation would improve readability.
5.For Experimental Results, the paper mentions syntax and logic testing of CQL files but lacks quantitative results, which could demonstrate their real-world effectiveness.
6.This paper identifies challenges such as manual conversion efforts and incomplete testing but does not fully explore their practical implications in Section 4.2.
Discussion of Limitations
7.Some figures (e.g., Figure 8) could be refined in order to be clearer, such as improving label contrast or simplifying complex flowcharts.
8.The conclusion could stress the study’s practical impact (e.g., supporting medical informatics education).
Author Response
Reviewer 2
Comments and Suggestions for Authors
- The abstract provides a comprehensive overview of the research’s objectives, methods, and outcomes. However, it should compare itself with existing methods to highlight the advantages and improvements. (e.g., evaluation metrics: accuracy).
Response: Thank you. We have added the following texts to highlight our contribution in the Results section of the Abstract:
~~
The unique contributions of our work are in two folds: 1) we created ontology-based machine-processable and reusable immunization recommendation rules; 2) we created and shared multiple formats of immunization recommendation rules publicly that can be a valuable resource for medical and medical informatics communities.
~~
- The innovation of this study should be emphasized more explicitly, especially how CQL and FHIR break limitations of traditional approaches in converting human-readable guidelines into machine-processable rules.
Response: Thank you for the suggestion. We have added the following texts to the Significance of the work.
~~
We used patients’ HL7 FHIR resources, which serve as a virtual layer between the underneath EMR/EHR systems and our CDSS rules. Therefore, no universal format of EHR schema is required as long as the parent data can be transformed into FHIR resource format. In CQL rules format, we used the CDSS Ontology and the coded value sets, most of the latter are from the National Library of Medicine Value Set Authority Center. These steps increased the standardization of the CQL rules and paved the way for reusability and shareability of the CQL rules. We would like to emphasize the innovation of our project, even though we leveraged the FHIR and CQL significantly.
~~
- The title of Section 2.1 "Vaccination Schedule Rules Converting Workflow," could be more concise. You could change it to "Rule Conversion Workflow".
Response: Thanks, we have updated the title according to your suggestion.
- In the Data Presentation part, while Table 1 summarizes the distribution of rules, additional tables could enhance the description of data collection and preprocessing. For instance, a table outlining the preprocessing standards for different formats (tabular, CQL, Excel) and validation would improve readability.
Response: Thank you for the suggestion. We have added Table 2 now to summarize the suggested content.
- For Experimental Results, the paper mentions syntax and logic testing of CQL files but lacks quantitative results, which could demonstrate their real-world effectiveness.
Response: Thank you for your suggestions. These tests started with a lot of failures, and the process of finding out the reasons is a mix of debugging and reformatting the CQL files. We have added the following texts to the Results section to address this suggestion:
~~
We successfully developed and tested 12 CQL rules/files, and the development of the remaining vaccine rules into CQL format is ongoing. Although testing CQL file syntax was not a one-step success, the process involved numerous failures, debugging, modifications, and retesting, all of which were iterated upon before all CQL files passed the syntax tests successfully. The patient FHIR resources that were used to test CQL files were prepared based on the CDSS rules in Excel format. We used positive cases (can trigger the CDSS rules) and negative cases (cannot trigger the CDSS rules) during the testing. All CQL files were tested successfully with patient FHIRE resources.
~~
- This paper identifies challenges such as manual conversion efforts and incomplete testing but does not fully explore their practical implications in Section 4.2.
Response: Thank you for your suggestion. We have added the following paragraph to the end of section 4.2---Challenges.
~~
The practical implications of these challenges suggest that converting the CDSS rules into a machine-processable format, despite being a long-standing challenge in the field, is one step closer to these rules being broadly adopted and ultimately benefiting patient care. Other steps, such as promotion, adoption, and implementation, are at different dimensions, and require different expertise and experience; however, these steps are as important as the conversion step in order to fulfill the potential of these rules. Meanwhile, we have to acknowledge that even if all conversions, promotions, adoptions, and implementations are conducted successfully, seamlessly, and perfectly, this will not guarantee positive outcomes during operation if we consider the very likely possibility of unavailable or inaccessible patient data needed to execute these rules successfully. Therefore, we have to recognize the differences between an ideal lab setting and day-to-day operations to execute these rules. We must also put the expectations within the right context of daily operation, with some tolerance or understanding if there are any failures.
~~
Discussion of Limitations
- Some figures (e.g., Figure 8) could be refined in order to be clearer, such as improving label contrast or simplifying complex flowcharts.
Response: In Figure 8, we intend to present two levels of charts to provide an overview of the entire chart (the upper portion) and an enlarged portion of the chart to provide more details on what is included in the chart. We revised the Figure title to make the intention more explicit.
~~
Figure 8. CDSS rules for MMR recommendations in chart format-a whole chart for an overview of a chart (top) and a partial chart to provide granular details (lower).
~~
8.The conclusion could stress the study’s practical impact (e.g., supporting medical informatics education).
Response: Thank you for your suggestion. We have added the following sentence into the Conclusion section:
~~Abstract-Conclusion: The publicly shared CDSS rules can be used for health and biomedical informatics education and training purposes, too.
Conclusion: The CDSS rule resource, which is the result of our effort, can be used by the health IT industry, clinical care institutions or practices, and medical informatics educators as operational CDSS rules or educational materials for teaching and training.
~~
Reviewer 3 Report
Comments and Suggestions for Authors
This is a very well written paper on an important topic. The methods and results are described in detailed. The discussion is balanced and detailed, and recognizes the limitations. As part of the future work, could the authors prescribe a format/method for specifying the vaccination schedules in the future to reduce or even eliminate the need for such conversion. While the paper addresses the challenge of translating the legacy schedules, perhaps the need for the same can be ameliorated by proper specification/design.
Author Response
Reviewer 3
Comments and Suggestions for Authors
This is a very well written paper on an important topic. The methods and results are described in detailed. The discussion is balanced and detailed, and recognizes the limitations. As part of the future work, could the authors prescribe a format/method for specifying the vaccination schedules in the future to reduce or even eliminate the need for such conversion. While the paper addresses the challenge of translating the legacy schedules, perhaps the need for the same can be ameliorated by proper specification/design.
Response: Thank you for your suggestion. We have added the following texts in the future work section.
~~
Meanwhile, we will provide more detailed documentation, along with this manuscript, so that future efforts on converting vaccination rules can be significantly more structured or diminished.
~~
Round 2
Reviewer 1 Report
Comments and Suggestions for Authors
In Figure 4, the designations Excel-15, CQL-12, Charts-19, Tabular-15 and Tabular--- 2 are not entirely clear. They should be explained in the text not far from Figure 4. It is necessary to add spaces before the dash (after the words). Why are there three dashes in Tabular--- 2?
Author Response
In Figure 4, the designations Excel-15, CQL-12, Charts-19, Tabular-15 and Tabular--- 2 are not entirely clear. They should be explained in the text not far from Figure 4. It is necessary to add spaces before the dash (after the words). Why are there three dashes in Tabular--- 2?
Response: Thank you for the suggestion. We have updated Figure 4 and its title. Please see the attached file for Figure 4, and here is the Figure title.
Figure 4. A more detailed workflow of rule conversion in various formats, including 15 Tabular files, 15 Excel files, 19 Charts, and 12 CQL files
Reviewer 2 Report
Comments and Suggestions for Authors
The authors considered my queries.
Author Response
Thank you, reviewer, for your comments and constructive suggestions, all of which are greatly appreciated and will help us improve our manuscript.
Xia on behalf of all coauthors